# *Sphagneticola Trilobata* (L.) Pruski (Asteraceae) Methanol Extract Induces Apoptosis in Leukemia Cells through Suppression of BCR/ABL

**DOI:** 10.3390/plants10050980

**Published:** 2021-05-14

**Authors:** Hoang Thanh Chi, Nguyen Thi Lien Thuong, Bui Thi Kim Ly

**Affiliations:** Faculty of Food Technology, Institute of Applied Technology, Thu Dau Mot University, Thu Dau Mot City 820000, Binh Duong Province, Vietnam; chiht@tdmu.edu.vn (H.T.C.); thuongntl@tdmu.edu.vn (N.T.L.T.)

**Keywords:** *Sphagneticola trilobata*, antileukemia, chronic myeloid leukemia (CML), BCR/ABL

## Abstract

We will study the effects of the methanol extract of *Sphagneticola trilobata* (L.) Pruski (Asteraceae) (MeST) on the growth of leukemia cells that may contain the *BCR/ABL* gene. This study also clarifies the mechanism of this effect on these cells. For this purpose, the cells harboring wild-type BCR/ABL, imatinib-resistant BCR/ABL (K562 and TCCYT315I), or Ba/F3 cells transfected with wild-type or mutant *BCR/ABL* genes were used. The results showed that MeST effectively inhibited the viability of leukemia cells in both a dose- and time-dependent manner. The effect of MeST seems to be more sensitive in the cells that carry imatinib-resistant BCR/ABL (especially the T315I BCR/ABL mutation) than those with wild-type BCR/ABL. Furthermore, we have demonstrated that the death caused by MeST is apoptosis and the treatment with MeST could suppress the expression of BCR/ABL, subsequently altering the downstream cascade of BCR/ABL such as AKT and MAPK signaling. In conclusion, MeST has been able to suppress the growth of leukemia cells harboring BCR/ABL. The mechanism of the anti-leukemic effect of MeST on cells harboring imatinib-resistant BCR/ABL mutations could be due to the disruption of the BCR/ABL oncoprotein signaling cascade.

## 1. Introduction

Chronic myeloid leukemia (CML) is a type of cancer found in the blood and bone marrow. This disease is characterized by a reciprocal t(9;22) chromosomal translocation, resulting in the formation of the Philadelphia (Ph) chromosome containing the *BCR/ABL* gene [1]. CML patients have the benefit of treatment due to the introduction of imatinib, a tyrosine kinase inhibitor (TKI) [2]. However, the imatinib resistance emerged as a serious problem. As least four generations of TKIs have been developed and shown to be potentially effective in treatment [3]. However, not all the TKI resistance problems are solved.

There are a large number of studies focusing on the mechanism of resistance. One of the reasons is the mutations on the sequences of *BCR/ABL* [3]. Moreover, the mechanism of resistance also counts for some, such as the genomic amplification of *BCR/ABL* [4]. Focusing on blocking the activation of BCR/ABL is very effective as imatinib has done, but it may not be the most perfect solution. A combination of reagents that have different effects of action may solve the problem of drug resistance [5].

Traditional medicine, focusing on herbal medicine with healthy activities, received largely concerns from researchers. The anticancer effect of herbal medicine is shown in many researches, but the mechanism of anticancer effect from it is difficult to clarify. It could be the combination of a variety of actions [6]. Many plant-derived medicines have been used in the community for a long time, demonstrating their safety in part. Therefore, it is a potential resource for humans to find effective therapeutic reagents for treatment. In fact, certain cancer chemotherapeutic drugs obtained from natural origin have had excellent effects [7]. The roles of traditional medicine have been established in Vietnam and are progressively contributing to supporting the healthcare system as an alternative treatment. There are more than 12,000 plants and no less than 2500 species used in ethnomedicine [8,9], but few Vietnamese plants have been studied for cancer [10,11,12].

*Sphagneticola trilobata* (L.) Pruski is a species of the genus Sphagneticola (Asteraceae family) and has many synonyms, but the most common one is *Wedelia trilobata* (L.) Hitchc. *S.trilobata* native to South America, Central America, Mexico and the West Indies. It has been widely found in Bangladesh, India, China, Malaysia, Indonesia, Vietnam, Cambodia and Myanmar. It thrives in valleys, ditches, wet roads, cropland, natural forests, pastures, coastal areas and urban areas. In Vietnam, *S.trilobata* is often referred to as Sai dat ba thuy, Sai dat kieng, Son cuc ba thuy, Cuc xuyen chi. It is a wild and highly invasive plant, but it is also cultivated and used by the Vietnamese as a traditional medicine. Previous studies state that the methanol flower extract of *S.trilobata* had excellent antioxidant activities [13]. The aqueous extracts of *S.trilobata* exerted a considerable hypoglycemic effect and had antidiabetic potential activity [14]. The methanol extract of the root, stem, leaves and flowers of *S. trilobata* was found to be significant against the following three bacteria: *Pseudomonas aeruginosa, Staphylococcus aureus* and *Salmonella typhi* [15]. *S.trilobata* methanol extract inhibited the growth of the human megakaryoblastic leukemia cell line MEG-01 with a half maximal inhibitory concentration (IC_50_) value of around 80 μg/mL [16]. Sesquiterpene lactone compound isolated from *S. trilobata* significantly inhibited the proliferation of HL-60, K-562, SI80, HepG2 and MCF-7 cells, with an IC_50_ value of 4 µg/mL, 6.5 µg/mL, 8 µg/mL, 24 µg/mL and 36 µg/mL, respectively [17].

In this study, we test the antileukemic effect of the methanol extract of *S.trilobata* (MeST) on CML cells harboring *BCR/ABL*. The results show that MeST has a growth inhibitory effect, not only in the imatinib-sensitive cells but in the imatinib-resistant cells. Moreover, we demonstrated that the MeST-induced cell death could be due to the disruption of the BCR/ABL signaling cascade in these cells.

## 2. Results

### 2.1. Effect of MeST on Cells Harbouring BCR/ABL Gene

In this study, we tested the effect of MeST on the growth of cells that carry the fusion gene *BCR/ABL* (wild-type or mutation). Cell lines were treated with different concentrations of extract for 2 days. Then trypan blue test was used to calculate the growth of the cell. The result showed that MeST could suppress the growth of human leukemic K562 cells as well as TCCY-T315I cells in a dose-dependent manner (Figure 1). The same effect could be observed on the transfected Ba/F3 cells (Ba/F3 cells were transfected with Y253H, E279K and T315I construct) (Figure 1). This observation is very interesting because MeST could overcome the imatinib resistance in cells that harbor T315I mutation in *BCR/ABL*.

To check whether the effect of MeST on cells is a time-dependent effect, we set up the time course experiment in TCCY-T315I cells. The result show that the cells treated with the control (DMSO alone) grow well after 48 h. However, the cells treated with 50 µg/mL of the MeST are not grown, and even undergo degradation, after 48 h of being cultured (Figure 2). This result again confirmed that MeST could suppress the growth of cells.

Since the SI value indicates the differential behavior of the extract, the more selective it is, the higher the SI value. An SI value greater than 3 units indicates the general toxicity of the pure compound [18]. In this study, the SI value was expressed as an IC_50_ value of the Vero cell compared to an IC_50_ value of the cancer cell line. As shown in Figure 3A, the effect of MeST on Vero cells is significantly less sensitive compared to other cells. The results showed that the SI values of MeST were higher than 3 (Table 1 and Figure 3B), indicating that the MeST had potent cytotoxic activity and good selectivity against leukemia cells with *BCR/ABL*.

### 2.2. Cell Morphological Changes Using Phase Contrast Inverted Microscope

Morphological changes are a sign of apoptosis [19,20]. In this assay, the morphological properties of the cells were determined by a phase contrast inverted microscope after exposure to different concentrations of MeST (50–100 µg/mL). As shown in Figure 4, all the treated cells are affected by MeST. The cells seem to be shrunken 24 h after being treated with 50 µg/mL or 100 µg/mL of MeST. This observation is compatible with the result shown in Figure 1. Changes in the cell morphology of TCCY-T315I were also found in both the treated and untreated cells (Figure 5).

These findings suggested that MeST could induce apoptosis on leukemia cells.

### 2.3. Apoptotic Cell Death Induced by MeST

The growth inhibition after treatment with MeST could involve changes in the cell morphology, as demonstrated in Figure 1, Figure 2, Figure 3 and Figure 4. In this experiment, we tested whether MeST treatment could also modify the expression of apoptotic markers. TCCY-T315I cells were treated with 50 µg/mL of MeST for up to 24 h. The results show that MeST treatment caused the activation of caspase. Briefly, we found that PARP and caspase-3 are cleaved in cells treated with 50 µg/mL of MeST (Figure 6A), and this result demonstrated that apoptosis happened. Next, we checked the nucleus morphology of cells. As shown in Figure 6B, the nucleus in cells treated with MeST is fragmented. In contrast, the nucleus is still intact in the control cells (untreated with MeST). Taken together, the cell death caused by MeST could be apoptosis.

### 2.4. MeST Could Suppress the Expression of BCR/ABL

BCR/ABL is considered as a targeted marker for treatment. Because of its significance, we tested whether cells treated with MeST could have an effect on the expression of BCR/ABL. K562, BaF3/T315I, and TCCY-T315I cells were treated with various concentrations of MeST for 24 h. As shown in Figure 7 (upper row), BCR/ABL expression is suppressed 24 h after treatment in a dose-dependent manner. Moreover, the downstream signaling of BCR/ABL, such as AKT and MAPK, is also affected and is compatible with the result of BCR/ABL suppression (Figure 7, middle row). This finding suggests that the antileukemia activity of MeST could act through the disruption of the BCR/ABL signaling cascade.

## 3. Discussion

Cancer is concerned as the main reason of death all over the world owing to late diagnosis, poor prognosis or drug resistance. It is estimated that death caused by cancer will dramatically increase in the next decades. Despite the excellent finding of tyrosine kinase-targeted therapy for cancer treatment, such as imatinib for CML with BCR/ABL [21] or erlotinib for non-small-cell lung carcinoma [22], the resistance to the treatment still remains a difficult problem and there is a need to overcome it to get better healthcare. Natural compounds from plants get much attention from researchers because of the potential source of reagents for screening suitable candidates for clinical use. In fact, a great number of clinically active drugs that are used in cancer therapy are either natural products or based on natural products [23]. In this report, we observed the growth inhibitory effect of MeST on leukemia cells and the induction of apoptosis in CML cells by MeST. So far, it has been reported that MeST inhibited the growth of the human megakaryoblastic leukemia cell line MEG-01 with an IC_50_ value of around 80 μg/mL [16]. The water extract of *S.trilobata* collected in Hongkong killed more than 90% of HL60 and K562 cells at a concentration of 200 µg/mL [17]. Thus, compared to previous studies, the results of our study show that *S.trilobata* collected in Vietnam is more sensitive to myeloid leukemia cells, with 100% of the cells having died after 48 h of being co-cultured with 100 µg/mL of MeST.

Apoptosis was based on some morphological changes in cell death including cell shrinkage, condensation of the chromatin, fragmentation of the nucleus, and blebbing of the membranes [24,25]. In this test, after two days of culturing, the treatment leukemia cells showed properties of apoptosis, such as cell shrinkage and membrane blebbing compared to untreated cells (Figure 1 and Figure 2). Cellular shrinkage is one of the leading morphological features of almost all apoptotic cell death due to abnormal changes in the intracellular water [26]. Moreover, checking the apoptotic markers after treating with MeST has shown that the PARP cleavage (89 kDa) and caspase-3 cleavage are present after treatment. The results of these experiments confirmed that MeST could possess the antileukemia effect through a mechanism that could reverse the cells from uncontrolled growth to programmed cell death. The results also showed that the higher concentration of MeST exposed to the leukemia cells, the more damaging the cell morphology changes compared to control cells.

We found in this report that the effect of MeST could modify the expression of BCR/ABL and therefore affect the activation of MAPK and AKT (Figure 7). It is well known that BCR/ABL is the oncoprotein and contributes to the development of leukemia. Based on this finding, the inhibitors of BCR/ABL are developed and successfully used in clinical treatment. However, other strategies of treatment have tried to suppress the expression of BCR/ABL as an alternative option to block the signals of BCR/ABL. In this report, we observed that BCR/ABL is suppressed when cells are treated with MeST. Subsequently, the downstream signals of the BCR/ABL cascade are also affected. This observation could provide the mechanism of actions that MeST have on leukemia cells.

It is worthy to notice that the MeST has very little effect on the Vero cells (the normal monkey kidney) compared to the leukemia cells (Figure 3A). When calculating the SI value of leukemia cells or transfected Ba/F3 cells with the oncogene BCR/ABL, we found that MeST has a high selective effect on abnormal cells (Figure 3B and Table 1).

## 4. Materials and Methods

### 4.1. Plant Materials and Sample Extraction Preparation

The fresh aerial parts of *S.trilobata* were collected in Hochiminh city (Vietnam) in 2017 and identified by Dr. My Van Dang, a herbalist of the Traditional Medicine Centre, Tinh Bien, An Giang province (the voucher number is HCM-2017-0114). The samples were washed and dried in a dry oven at 40 °C after collection until they got to a constant weight. The crude methanol extract was prepared as described in detail before [27]. The phytochemical analysis [28] showed that alkaloid, coumarin, flavonoid, anthocyanin, cardiac glycoside, tannin, reduced sugar, and polyuronide compounds have been seen in the *S.trilobata* extract.

### 4.2. Cell Lines, Culture Conditions

The human leukemia cell lines K562, TCCY-T315I and Ba/F3 were received from Prof. Yuko Sato (Tokyo, Japan). The Ba/F3 cells with T315I, Y253H and E279K were created as previous reported [20]. The African green monkey kidney (Vero) cell lineage (ATCC CCL-81™) was used in this study. Vero cells were grown in Dulbecco’s modified Eagle’s medium (DMEM, Sigma-Aldrich, Ho Chi Minh City, Vietnam) and other cells were grown in Roswell Park Memorial Institute 1640 medium (RPMI 1640, Sigma-Aldrich, Ho Chi Minh City, Vietnam) supplemented with 10% heat-inactivated fetal bovine serum (FBS) (ThermoFisher Scientific, Ho Chi Minh City, Vietnam), 100 IU/mL penicillin, and 0.1 mg/mL streptomycin (P4333, Sigma-Aldrich, Ho Chi Minh City, Vietnam) in a humidified incubator of 5% CO_2_ at 37 °C.

### 4.3. Cell Viability

Cell proliferation was conducted on suspension cells and was determined by the trypan blue dye exclusion test as described previously [21].

All of the cells (K562, TCCY-T315I, Ba/F3-T315I, Y253H, and E279K) that live on the signal induced by BCR–ABL overexpression were used in the research, while the un-transfected Ba/F3 cells were not used because their growth was dependent on the signal induced by IL3. The selectivity index (SI) was expressed as IC_50_ value of Vero cell/IC_50_ value of cancer cell line. The SI values indicate MeST selectively kill leukemia cells and not just non-selective cytotoxic extracts. Samples with an SI value >3 were considered highly selective for cancer cells [22].

### 4.4. Morphological Changes of Cells by Phase-Contrast Microscope

All cell lines excepted Vero cells were seeded in 6-well plates at a concentration of 105 cells per well and incubated overnight. Then, the cells were treated with different concentrations of MeST (0, 50 and 100 µg/mL) followed by 48-h or 72-h incubation at 37 °C with 5% CO_2_. The untreated cells served as a control. The morphological changes of the cells were observed using an inverted light microscope at 10× magnification.

### 4.5. Western Blot Analysis

Cells were plated onto 10 cm dishes at a density of 1 × 10^5^ cells/mL in the presence of various concentrations of reagents. After incubation for indicated durations, cells were collected and washed twice with PBS (−). Cell protein was extracted and Western blot analysis was done as described previously [23]. A c-Abl (sc-23) antibody was obtained from Santa Cruz Biotechnology (Santa Cruz, CA, USA). Anti-actin (A2066) was obtained from Sigma-Aldrich. Then, p44/42 MAPK (Erk1/2), phospho-p44/42 MAPK (Thr202/Tyr204), AKT, phospho-AKT (Ser473), and caspase-3 antibodies were from Cell Signaling Technology Japan (Tokyo, Japan). Anti-PARP antibody was from WAKO Chemicals (Osaka, Japan).

### 4.6. Statistics

The data were analyzed using the unpaired Student’s t-test between the control and compounds. A *p* value <0.05 was considered statistically significant. Data were compiled from three independent experiments and values were expressed as mean ± SD. For data calculations GraphPad prism software, version 8.3.0, (GraphPad Software Inc. HoChiMinh City, Vietnam) was used.

## 5. Conclusions

We propose that MeST could serve as a plant candidate for leukemia treatment. However, more studies need to be conduct in order to apply the benefit of MeST in practice.

## Figures and Tables

**Figure 1 plants-10-00980-f001:**
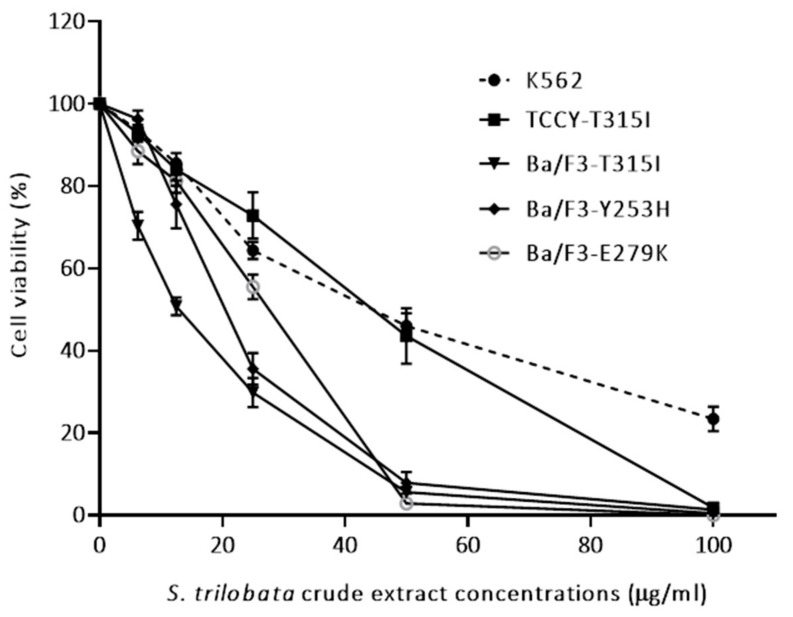
Cell growth inhibition induced by MeST in a dose-dependent manner. MeST were added at different concentrations (6.25 to 100 µg/mL) to K562, TCCY-T315I, Ba/F3-Y253H, Ba/F3-E279K, and Ba/F3-T315I cells for 48 h and cell viability assays were performed as described in materials and methods. Results are presented as a percentage of death from treated MeST cells in comparison with non-treated MeST cells (handled with DMSO at the same concentration as MeST). The experiments were performed in triplicate. *p* value < 0.0001 for all tested cells.

**Figure 2 plants-10-00980-f002:**
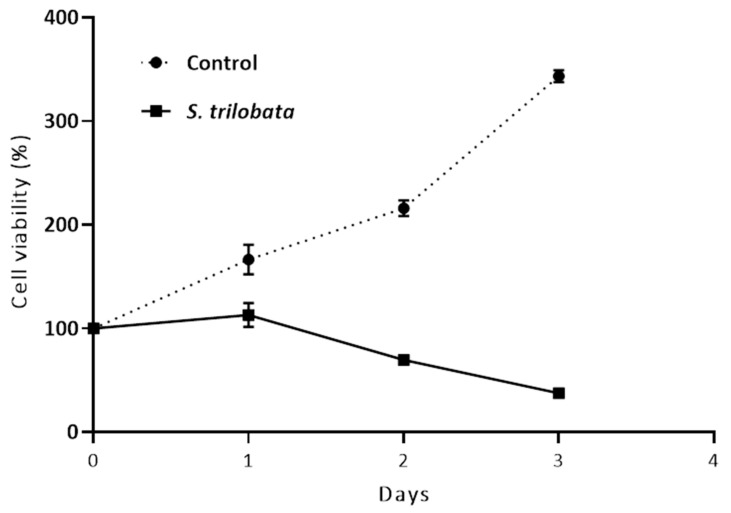
Effect of MeST on TCCY-T315I cells is a time-dependent manner. MeST (50  µg/mL) were added to TCCY-T315I cells for various time periods (0 to 3 days) and cell viability assays were determined as described in materials and methods (*p* < 0.0001).

**Figure 3 plants-10-00980-f003:**
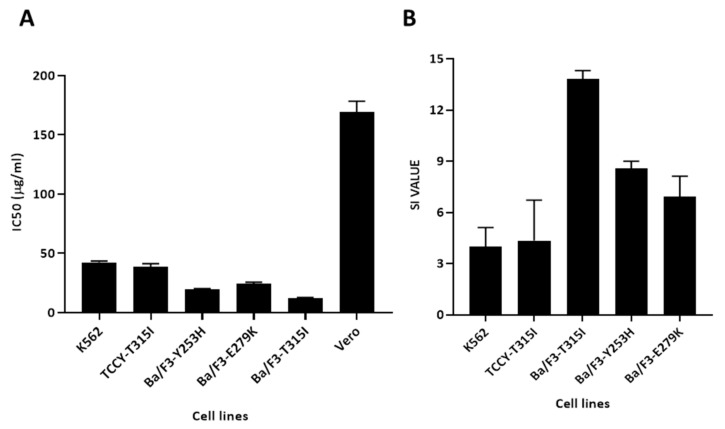
The IC_50_ and SI values of MeST on cells. (**A**) The IC_50_ value of MeST on K562, TCCY-T315I, Ba/F3-T315I, Ba/F3-Y253H, Ba/F3-E279K, and Vero cell. (**B**) The SI value of MeST on K562, TCCY-T315I, Ba/F3-T315I, Ba/F3-Y253H, Ba/F3-E279K were calculate by dividing IC_50_ against Vero cells to IC_50_ against leukemia cells.

**Figure 4 plants-10-00980-f004:**
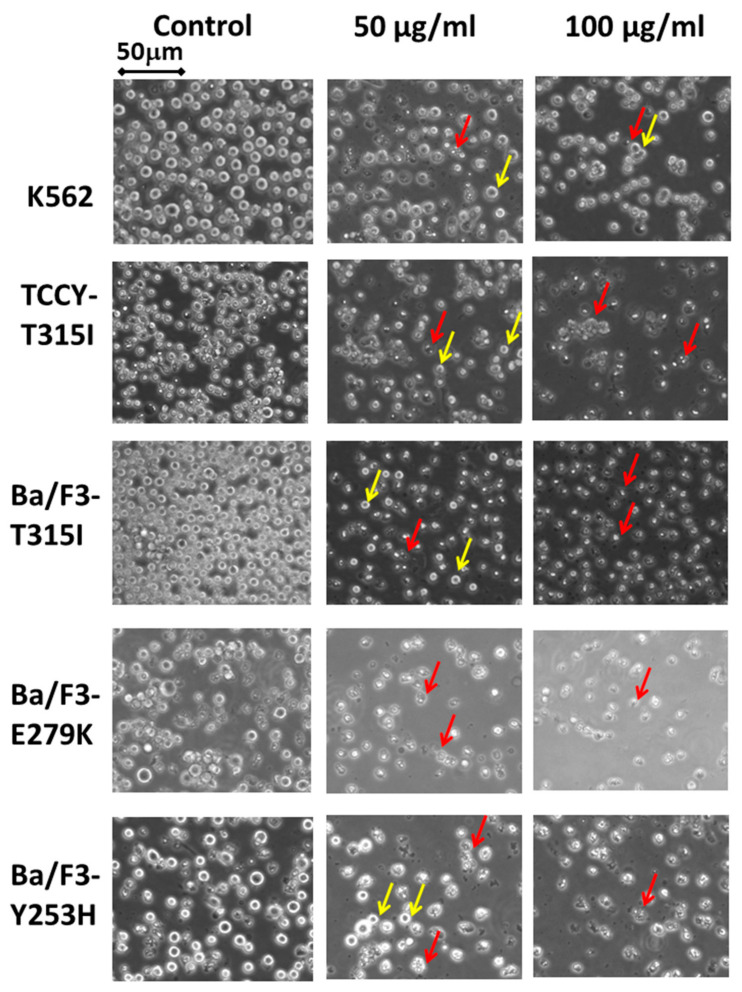
Morphological changes in cells by phase contrast microscope. K562, TCCY-T315I, Ba/F3-T315I, Ba/F3-E279K and Ba/F3-Y253H cells were treated with 0, 50 or 100 µg/mL of MeST for 48 h and then observed under a phase contrast inverted microscope. (Intact cells are shown by a yellow arrow, debris cells are indicated by a red arrow.)

**Figure 5 plants-10-00980-f005:**
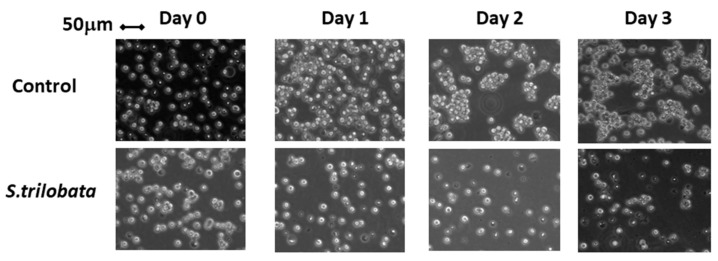
Morphological changes of TCCY-T315I by phase contrast microscope. TCCY-T315I cells were treated with 50 µg/mL of MeST for 24, 48 and 72 h and then observed under phase contrast inverted microscope.

**Figure 6 plants-10-00980-f006:**
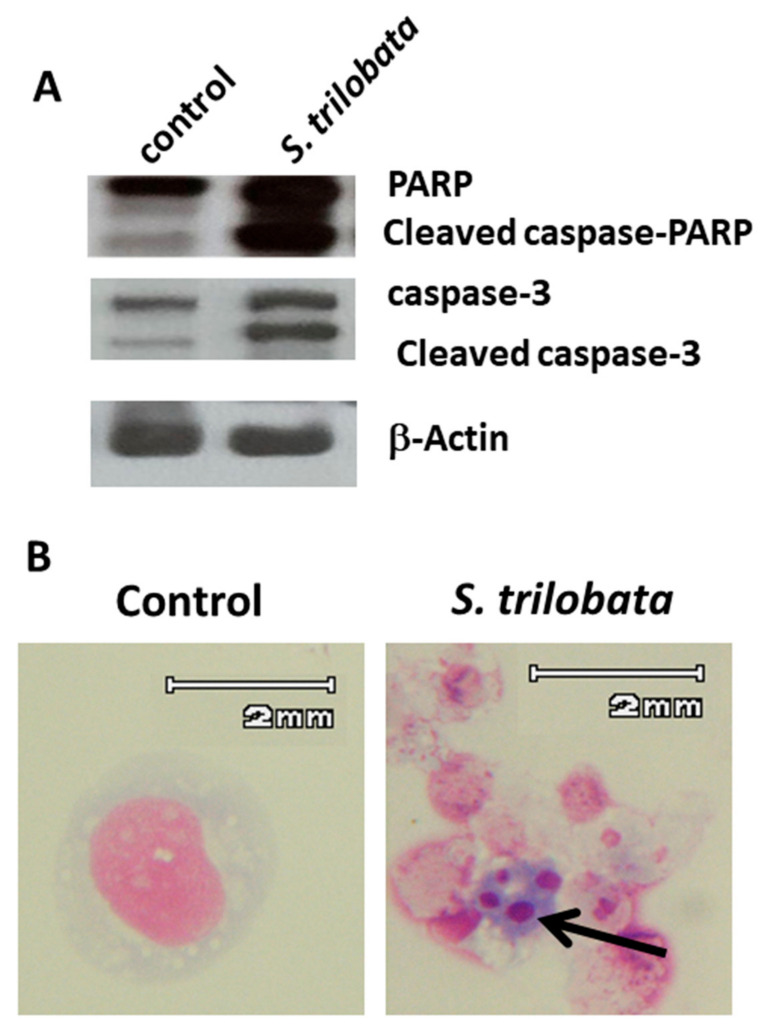
MeST could induce apoptosis in cells. (**A**) TCCY-T315I cells were treated with 50 µg/mL of MeST for up to 24 h. Cell lysates were subjected to Western blot analysis with caspase-3 and PARP antibody. The results showed that treatment with MeST induced the cleavage of caspase-3 and PARP. (**B**) The morphological changes of TCCY-T315I cells by MeST treatment. The arrow showed the appearance of apoptotic body in TCCY-T315I cells after 24 h treating with MeST.

**Figure 7 plants-10-00980-f007:**
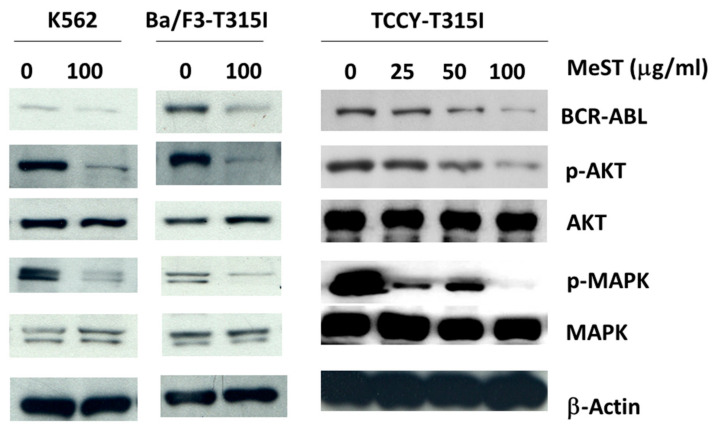
MeST could suppress the expression of BCR/ABL in cells. K562, BaF3/T315I, and TCCY-T315I were treated with indicated concentrations of MeST or DMSO alone as a control. Total cell lysates were subjected to Western blot analysis with indicated antibodies.

**Table 1 plants-10-00980-t001:** The IC_50_ value and selectivity index of MeST on cell growth of leukemia cell lines.

Leukemia Cell Lines	MeST	
	IC_50_ (µg/mL)	SI
K562	42.31 ± 1.13	4.00
TCCY-T315I	38.78 ± 2.37	4.36
Ba/F3-T315I	12.23 ± 0.51	13.81
Ba/F3-Y253H	19.68 ± 0.41	8.59
Ba/F3-E279K	24.41 ± 1.21	6.92

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
