# Peer review of "Sphagneticola Trilobata (L.) Pruski (Asteraceae) Methanol Extract Induces Apoptosis in Leukemia Cells through Suppression of BCR/ABL"

_plants, 2021, doi:10.3390/plants10050980_

Round 1
Reviewer 1 Report
Dear Authors. The submitted The MS “Sphagneticola trilobata (L.) Pruski (Asteraceae) methanol ex-tract induces apoptosis in leukemia cells through suppression of BCR/ABL. By Hoang Thanh Chi, Nguyen Thi Lien Thuong, and Bui Thi Kim Ly.” For possible publication at the Journal. In the abstract, the phrase “This research was planned to test the effects of Sphagneticola trilobata (L.) Pruski (Asteraceae) methanol extract on the growth of leukemia cells that harboring the BCR/ABL gene.” Change by “The effects of the methanol extract of Sphagneticola trilobata (L.) Pruski (Asteraceae) on the growth of leukemia cells that may contain the BCR/ABL gene will study.” In Introduction say: Chronic myeloid leukemia is a type of cancer found in the blood and bone marrow, characterized by a reciprocal chromosomal translocation, resulting in the formation of the Philadelphia (Ph) chromosome containing the BCR/ABL gene, the patients has been treated with imatinib, a tyrosine kinase inhibitor, but not all the resistance problems to it are solved. These could be the mutations on the sequences of BCR/ABL or on genomic amplification of BCR/ABL. Here, tested the ant leukemic effect of methanol extract S. trilobata on leukemia cells that may contain the BCR/ABL gene. The results showed that the methanol extract inhibited the viability of leukemia cells in both dose and time-dependent manner, being more sensitive in the cells carrying imatinib resistant BCR/ABL gene (especially T315I BCR/ABL mutation) than the wild type BCR/ABL. They demonstrated that the death caused by S.trilobata extract is apoptosis and the extract could suppress the expression of BCR/ABL gene, by altering the downstream cascade of BCR/ABL such as AKT and MAPK signaling. Therefore, they proposed the mechanism of ant leukemic effect of S.trilobata on imatinib resistant BCR/ABL mutations cells could be due to the disruption of BCR/ABL onco protein signaling cascade. I suggest to use “methanol extract of S.trilobata” than “extract or S.trilobata extract or methanol extract in all the MS”. Indicate the concentration like 80 or 100 mg/ml methanol extract of S.trilobata that inhibited 100 % the cell growth (Fig. 1). Change “methanolic” by “methanol”. In conclusion change the phrase “This finding suggest that S.trilobata could serve as a plant candidate for leukemia treatment” by “We propose that methanol extract of S. trilobata could serve as a plant candidate for leukemia treatment”
Reviewer 2 Report
General comments
This study by Chi et al., shows cellular apoptosis induced by methanol extract of Sphagneticola trilobata. Extract treatment of leukemia cell lines and kidney-derived cell lines, coupled with western blot analyses, convincingly demonstrate that the growth inhibitory effects of S.trilobata extract are mediated via disruption of BCR/ABL signaling.
However, the authors’ claim of an anti-leukemic effect of S.trilobata extract, is not substantiated with the appropriate negative controls. Extract-treated non-cancerous bone-marrow derived cells or cell lines, with no inhibitory effect on cell growth will be an indication that the apoptotic effect of the extract is specific to cancerous bone marrow derived cells or cell lines.
At best, this study describes a growth inhibitory mechanism for methanol extracted S.trilobata extract. There is a general discrepancy between what is actually shown (induction of apoptosis) and what is claimed (anti-cancer effect)
In their discussions, they also cited similar studies where the growth inhibitory properties of differently extracted S. trilobata was studied. By not including, these different 'types' of S. trilobata in their own study for side-by-side comparisons, I find that the novelty of the current study is quite low.
Specific comments
- Figure 1: Untransfected control of Ba/F3 cells have not been included
- Figure 3: What is the rational for using normal kidney cells from a different species (monkey) as non-cancer control to human bone marrow derived cancer cells?
- Figure 4: The chosen microscopic method and resolution, does not show differences in cell morphology. From left to right, changes in cell numbers, plus debris from dead cells are seen. Not morphological changes
- Figure 6B: the arrow is in the wrong image
Reviewer 3 Report
The manuscript “Sphagneticola trilobata (L.) Pruski (Asteraceae) methanol extract induces apoptosis in leukaemia cells through suppression of BCR/ABL. by Chi et al. presents the results on the effect of S. trilobata extract on viability and cell death of selected leukaemia cells. The author declared that S. trilobata extract has been able to suppress the growth of leukaemia cells harbouring BCR/ABL. They also proposed that the mechanism of anti-leukemic effect of S. trilobata on cells harbouring imatinib resistant BCR/ABL mutations could be due to the disruption of BCR/ABL oncoprotein signalling cascade. However, they do not provide the solid evidence to support this.
Overall, the manuscript suffers the inconsistency of the results. Most importantly, the positive/negative controls to prove the specific effect of S. trilobata extract were not included.
Major comments:
The composition of S. trilobata extract was not analysed and presented.
Lane 46: The sentence “However, all of the herbal medicine are safe for human due to the long-time use in the population” should be soften. Actually, some herbal medicine may have the adverse health effects.
Lane 124: The authors wrote: Different concentrations of S. trilobata extract were treated with cell lines for 2 days. This should read “Cell lines were treated with different concentrations of extract for 2 days.”
Figure 1: Effects of imatinib, or other TKIs on the viability of tested cell lines should be included (mandatory).
Lane 143: Why the DMSO was used as a vehicle. In the Figure 1 the experiment was performed using MeOH (lane 138).
Lane 173: “Changes in cell morphology of TCCY-T315I were also found in both treated and untreated cells (Figure 5).” What is the reason that morphology of TCCY-T315I was affected? This should be explained.
Figure 6A: Specific inhibitors of caspases should be used in combination with the S. trilobata extract (e.g. z-IETD-fmk, z-DEVD-fmk and z-LEHD-fmk) to show that the cell death induced by the tested extract is caspase specific (mandatory). This should be correlated with the proliferation and viability studies.
Figure 6: The authors declared that “The arrow showed the appearance of apoptotic body in TCCY-T315I cells after 24 hrs treating with S. trilobata extract.” Actually, the arrow is pointing on control cells.
Figure 7: The effect of S. trilobata extract on the level of BCR-ABL, AKT and MAPK should be evaluated for all cell lines, including the K562 cells.
The potential of S. trilobata extract to induce apoptosis in the K562, TCCY-T315I and Ba/F3 (T315I, Y253H and E279K) must be evaluated by Annexin-V/PI assay to prove that S. trilobata extract is active also against imatinib resistant cells (mandatory). The viability/proliferation assay showing decreased proliferation/viability of treated cells is not sufficient.
Reviewer 4 Report
Manuscript title: Sphagneticola trilobata (L.) Pruski (Asteraceae) methanol extract induces apoptosis in leukemia cells through suppression of BCR/ABL
Comments
- Authors have mentioned "S.trilobata" throughout the manuscript with out an space after S. It should be corrected as S. trilobata.
- Line 61:Correct to "Previous 61 studies state that"
- The labels in figures are not clear. I suggest authors to use Arial or Calbiri inside figures so that they will be more readable.
- Figre 3 and other places: IC50 ; 50 should be in subscript.
- Line 182: change to extract
- Figure 7: Please mention the unit of concentration of extracts (ug/ml)
- Although the bioassay section of manuscript is well designed and presented, this manuscript lacks the phytochemical aspect. I understand that authors have screened the extract for its activity, however, they should at least provide chemical profiles of the extract using HPLC or LC-MS. This is not only important to know about the possible chemical presents in extract, but will also be important in reproducibility of the research results. The type and amount of any phytochemical present in extract will depend upon many factors such as extraction condition, solvent, plant collection season, etc. Providing chemical profile will help to know the consistency of the extracts in future.
Round 2
Reviewer 2 Report
The authors have made some minor improvements to the manuscript.
My updated comments are indicated below their response to my previous concerns:
Point 1: Figure 1: Untransfected control of Ba/F3 cells have not been included
Response 1: In this experiment, the untransfected Ba/F3 cells live dependent on signal induced by IL3 whereas all the tested cell lines live depend on signal induced by overexpression of BCR-ABL sothat untransfected Ba/F3 cells are notsuitable as the control in this case.
Reviewer: Please update methods section, to inform on the rationale for each of the used cell lines.
Point 2:Figure 3: What is the rational for using normal kidney cells from a different species (monkey) as non-cancer control to human bone marrow derived cancer cells?
Response 2:It is preferable to use normal bone marrow cells as a control. However, in our situation, normal kidney cells are the better option.
Reviewer: Authors have not provided any scientific justification
Point 3:Figure 4: The chosen microscopic method and resolution, does not show differences in cell morphology. From left to right, changes in cell numbers, plus debris from dead cells are seen. Not morphological changes
Response 2:In this experiment, we want to demonstrate that morphology cell modifications exist in the population of control cells as compared to treated cells, rather than just one or two cells.
Reviewer: As presented, the figures show increasing and decreasing cell numbers, as well as cellular debris. The authors could use arrows to magnify specific sections of the figures to show clearly cell morphology changes in a few cells.
Point 4:Figure 6B: the arrow is in the wrong image
Response 2:We've fixed it, as seen below.
Reviewer: Satisfied
Reviewer 3 Report
-
Author Response
The reviewer gave no further comments.
Reviewer 4 Report
Authors have revised as suggested.
In figures 5, 6, and 7, the plant name is written as S. tribolata which should be corrected to S. trilobata.
As authors wrote in reply, that their manuscript on Chemical isolation has already been accepted in JBUON, I recommend to add the results of that study in few sentences. This can be cited as in press.
Round 3
Reviewer 2 Report
Requested modifications have been made. I have no further comments
Reviewer 4 Report
Authors have revised manuscript as suggested.
Author Response
There are no comments from the reviewer.